



# Impacts of anemometer changes, site relocations and
# processing methods on wind speed trends in China
Yi Liu[1], Lihong Zhou[1], Yingzuo Qin[1], Cesar Azorin-Molina[2], Cheng Shen[3], Rongrong
Xu[1*], Zhenzhong Zeng[1*]
[1] School of Environmental Science and Engineering, Southern University of Science and Technology,
Shenzhen, China
[2] Centro de Investigaciones sobre Desertificación, Consejo Superior de Investigaciones Científicas
(CIDE, CSIC-UV-Generalitat Valenciana), Climate, Atmosphere and Ocean Laboratory (Climatoc-Lab),
Moncada, Valencia, Spain
[3] Regional Climate Group, Department of Earth Sciences, University of Gothenburg, Gothenburg,
Sweden
*Correspondence: xurr@sustech.edu.cn (R. X); zengzz@sustech.edu.cn (Z. Zeng)
Manuscript for *Atmospheric Measurement Techniques*
March 23, 2023





## Abstract

*In-situ* surface wind observation is a critical meteorological data source for various research fields. However, data quality is affected by factors such as surface friction changes, station relocations, and anemometer updates. Previous methods to address discontinuities have been insufficient, and processing methods have not always adhered to World Meteorological Organization (WMO) guidelines. We analyzed data discontinuity caused by anemometer changes and station relocations in China's daily *in-situ* near-surface (~10m) wind speed observations and the impact of the processing methods on wind speed trends. By comparing the wind speed discontinuities with the recorded location changes, we identified 90 stations that showed abnormally increasing wind speeds due to relocation. After removing those stations, we followed a standard quality control method recommended by the World Meteorological Organization to improve the data reliability and applied Thiessen Polygons to calculate the area-weighted average wind speed. The result shows that China's recent reversal of wind speed was reduced by 41% after removing the problematic stations, with an increasing trend of 0.017 m s$^{-1}$ year$^{-1}$ (R$^2$ = 0.64, P < 0.05), emphasizing the importance of robust quality control and homogenization protocols in wind trend assessments.

**Keywords.** wind speed trends; anemometer changes; station relocations; processing methods; quality control; homogenization





## 1. Introduction

*In-situ* surface wind observation is a key meteorological data that has been used in various avenues of research, e.g., wind power evaluation (Tian et al., 2019; Zeng et al., 2019; Liu et al. 2022a), extreme wind hazard monitoring and prevention (Zhou et al., 2002; Tamura, 2009; Liu et al., 2022b), and evapotranspiration analysis (Rayner, 2007; McVicar et al., 2012), to name but a few. The application of robust quality control and homogenization protocols are crucial for generating reliable wind speed time series for further trend and variability analyses (Azorin-Molina et al., 2014; Azorin-Molina et al., 2019).

Wind data quality is affected by surrounding surface friction change, station location issues, and anemometer changes in type and height (Masters et al., 2010; Wan et al., 2010; Cao & Yan, 2012; Hong et al., 2014; He et al., 2014; Azorin-Molina et al., 2018; Camuffo et al., 2020). Surrounding surface friction changes are mainly associated with urbanization (Zhang et al., 2022) and vegetation growth (Vautard et al., 2010), which modify wind speed fields around the stations. Because of these issues, stations are relocated to satisfy observing criteria (Trewin, 2010). Station relocation is quite common in rapidly developing countries. For instance, about 60% of stations in China experienced relocation (Sohu, 2004). Some relocation-caused breakpoints have been corrected by parallel observations (i.e. operating observations for an overlapping period at both the old and new observing stations; CMA, 2011; CMA, 2012; WMO, 2020), but not all (Feng et al., 2004; Fu et al., 2011; Patzert et al., 2016; Tian et al., 2019; Yang et al., 2021). Besides relocation caused by rapid urbanization (or vegetation growth), updates to automatic anemographs at the beginning of the 21$^{st}$ century in China also caused discontinuities in wind series (Fu et al., 2011).

Scientists have tried different methods to handle discontinuities. Tian et al. (2019) and Yang et al. (2021) deleted stations with recorded changes in latitudes, longitudes or altitudes, but they omitted to check whether those recorded relocations caused an abrupt discontinuity in the time series or if parallel observations have corrected them. This results in some stations being mistakenly deleted and significantly reduced the number





of available stations. Other research used statistical methods to detect or correct the
time series' abnormal breakpoint (Feng et al., 2004; Wang, 2008). However, without
examining the causes behind the discontinuity, this may also mistakenly delete stations
with natural abrupt climatic changes (Bathiany et al., 2003). Combining those two
methods by matching discontinuity with recorded station relocation is needed. Li et al.
(2018) have manually checked the station histories for nine stations in North West
China, but an algorithm is required to apply this approach to large datasets.

Besides data discontinuities, the processing method also affects the wind series.

There are two critical steps in the processing: 1) selecting qualified stations and 2)
calculating the average value. As for the first step, World Meteorological Organization
(WMO) suggests deleting stations with either too much missing data or continuous
missing data (WMO, 2017). Previous studies only constrained the number of missing
values monthly (Zeng et al., 2019), yearly (Tian et al., 2019) or even in the whole period
(Yang et al., 2021) but didn't check whether the missing values were continuous. As for
the second step, most studies used the station average as the mean wind speed (Li et al.,
2017; Zeng et al., 2019; Tian et al., 2019; Yang et al., 2021; Shen et al., 2021; Zha et
al., 2021). However, station distribution and wind speed spatial variation are often
inhomogeneous (Feng et al., 2004; Fu et al., 2011; Liu et al., 2019). Therefore the
station average will have spatial biases. An improved average method to rearrange the
weight for each station, e.g. Thiessen Polygon (Fu et al., 2011), is needed.

Herein, taking stations in China as an example, we analyzed the existing data

discontinuities and their potential causes. Furthermore, we propose an improved
solution by using an algorithm to compare the statistic breakpoint with the recorded
relocation to double-check the discontinuity caused by relocation. Then using WMO's
quality control criteria and Thiessen Polygon (Thiessen, 1911), we generated wind
speed time series without temporal bias caused by heterogeneous missing values and
spatial biases caused by uneven station distribution.

**2. Dataset and methodology**

## 2.1 WMO quality control method

We used the China Surface Climatic Data Daily Data Set (CSD) (Version 3.0) from the China Meteorological Data Service Center (http://data.cma.cn/en/?r=data/; last accessed March 2020). The quality control method is recommended by WMO (2017), which required the following criteria before using the daily mean values in a month as monthly mean values: (1) <11 missing daily values in a month and (2) <5 consecutive missing daily values in a month. Qualifying stations must have monthly values for every month during the study period; Otherwise, the station will be completely excluded from the calculation. The station excluded by each criterion can be found in Table S1.

## 2.2 Station location changes in record

CSD provides daily wind speed and location information for 840 stations for 1961-2019. But there are some mistakes in the daily location records. For example, if the station location changed from A to B and back to A within a month, B is potentially a mistaken record. Therefore, we first use mode (the statistic term meaning the value that appears most often, here referring to the location with the highest frequency in a month) to resample the daily location to the monthly location. Second, considering that recorded longitude and latitude has the same spatial resolution of minutes, we defined the threshold of location change as the minimum accuracy of the longitude and latitude record, i.e., one minute. That is 1.85km for longitude and 1.85km*cosφ for latitude, where φ is the latitude. Third, as for altitude, we allow a 20m measuring error following Tian et al. (2019). A station with more than 20m change in altitude will be considered as relocation. It is noteworthy that CSD labels uncertain altitude records by adding 10km to the raw data (CMA, 2017), which are considered as no observations in our analysis. This way, we identified 432 stations as relocations from the 601 qualified stations after applying the WMO quality control (details in Table S2).

## 2.3 Breakpoint detection and the comparison with recorded relocation

We used Pruned Exact Linear Time (PELT) method (Killick, Fearnhead & Eckley,

2012) to detect the jumps in the mean level in the monthly wind speed time series (Fig
4a, Fig 4c). This method is a wrapped function named *findchangepts* in Matlab. PELT
is essentially a traversing method. For a time series with N values $(x_1, x_2 \ldots x_N)$, the
function uses equations 1 & 2 to calculate the total residual errors (J) for each point (k)
assumed as a breakpoint. The point with the most significant change in the mean (lowest
total residual errors, $J$) is reported as the breakpoint. The breakpoints here can be caused
by artificial relocations or natural climate changes.
$$J(k) = \sum_{i=1}^{k-1}\left(x_i - mean([x_1 \cdots x_{k-1}])\right)^2 + \sum_{i=k}^{N}\left(x_i - mean([x_k \cdots x_N])\right)^2 \quad (1)$$

$$mean([x_m \cdots x_n]) = \frac{1}{n-m+1}\sum_{r=m}^{n} x_r \quad (2)$$

Then we use relocation records to separate changes brought by artificial relocation

from changes in natural climate. If the breakpoint and one of the relocation dates (some
stations have more than one relocation record) happened in the same two months, we
will consider that the time series is significantly affected by the relocation, and the
station will be deleted. Stations with natural-climate-caused location changes will be
reserved.

The change point in the trend of the annual national average wind speed (Fig 2b,

Fig 5b) is detected following the method used by Wang et al. (2011). All the trends
reported are based on the least square fits.

**3. Results and discussion**
**3.1 Data issue due to anemometer changes**

We found a clear decline in the frequency of zeros in most CSD stations between

2002 and 2007 (Figure 1a), from 10-14 days per year to less than two days per year.
This clear drop is not a result of wrongly taking zero values as no observations (NaN)
as happened in the Integrated Surface Dataset (ISD, Dunn et al., 2022), as no abrupt
increase in NaN frequency was observed (Supplementary Figure S1). Instead, the
decline is accompanied by an improvement in record accuracy: i.e., the measurement



intervals became narrow (from 0, 0.3, 0.5, 0.7, 0.8, 1.0 m s⁻¹, etc. to 0, 0.1, 0.2, 0.3 m
s⁻¹, etc.; Figure 1b and Supplementary Figure S2). Taking Station Naomaohu in
Xinjiang (station ID: 57432) as an example, from 2002 to 2003, zero values decreased
from more than 30 days per year to less than five days per year and wind speed records
changed from 0, 0.3, 0.7, 1.0 m s⁻¹, etc. to 0, 0.3, 0.5, 0.8, 1.0 m s⁻¹, etc. Since 2004, the
record accuracy was further improved to 0.1, 0.2, 0.3… m s⁻¹ and zeros values almost
disappeared (Figure 1b).
This change is potentially caused by the transformation in measure frequency,
anemometer type and data logging, based on the station history recorded by Xin et al.
(2012). As for measurement frequency, in 2003, Station Naomaohu changed from 3
observations per day (i.e., 8:00, 14:00 and 20:00, China Standard Time) to four times
per day (2:00, 8:00, 14:00 and 20:00, China Standard Time). The increase in the
frequency of measurements decreases zeros in daily wind data, as only if all
observations report zero wind speeds, will the daily data (i.e., the average of all
observations in a day; CMA, 2017) be recorded as zero. Then in 2005, the EL contact
anemograph (Yang, 1986; Jin, 2011; Xin et al., 2012, Zhang et al., 2020) requiring
manual recording was changed to EC photoelectric encoder self-recording type (Kuang,
2016; Jin, 2011; Xin et al., 2012). Both EL and EC type anemographs use cup
anemometers to measure wind speed. This anemograph change further decreases the
likelihood of recording zero daily wind speed because the updated new anemometers
are more sensitive, and even very low wind speeds will be measured with a value
instead of recorded as zero (Azorin-Molina et al., 2018). The smooth increasing
frequency of zero values from 1960 until 2000 also supports this statement (Figure 1a):
the longer the anemometer is used, the less sensitive it will become, and hence a greater
wind speed will be required to record a non-zero value (Azorin-Molina et al., 2018),
overall increasing the zero values. As for the change in data accuracy, there are two
reasons: 1) EL type anemograph only measures the times of electronic contact (200
meters rotation distance per contact) in 10 mins, therefore it has discrete records. For
example, one contact means 0.3 m s⁻¹ (200m/600s) and two contacts means 0.7 m s⁻¹



(400m/600s) (Hu et al., 2009) while EC type has more accurate records using the Grey
Code; 2) the data logging changed from manual reading, calculating and rounding to
instrument automatically calculating and retaining one decimal place.

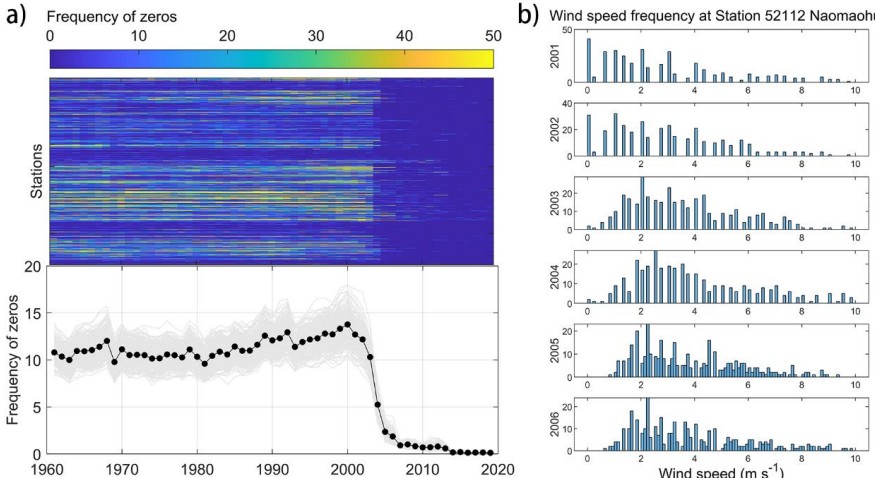


**Figure 1. Changes in wind speed data caused by anemoter updates. a)** Decrease of
frequency of zeros. Each horizontal bar in the upper figure represents one station and
there are 840 stations in total. The color indicates the frequency of zeros (days per year).
The black dotted line in the lower figure is the average annual frequency of zeros of all
the stations. The 300 grey lines are sample averages, each containing 40% amount of
the total stations. **b)** Frequency (days per year) of daily wind speed measurements
between 2001 and 2006 for Station #52112 Naomaohu (43°45′N, 94°59′E, 479.0 m
a.s.l.)

**3.2 Quality-controlled series**
Following WMO's criteria, we generated the monthly average wind speed for each
station (Figure 2a). We found that since January 2016, there have been 126 stations that
no longer have records (distribution see Figure S3). We compared the time series with
and without these stations and found the difference is not significant (t-test P < 0.001,
Figure 2b). To obtain a longer time series including recent years' data, we deleted the
126 stations and only used the 601 stations with complete monthly average wind speeds



for 1980-2019. The breakpoint was detected in 2011 (P < 0.001) with a decreasing trend
of -0.011 m s$^{-1}$ year$^{-1}$ (R$^2$ = 0.84, P < 0.001) before the breakpoint and an increasing
trend of +0.022 m s$^{-1}$ year$^{-1}$ (R$^2$ = 0.87, P < 0.001) after.




**Figure 2. Monthly average wind speed after being filtered by WMO's criteria. a)**
Each horizontal bar represents one station. Months with no data (NaNs) are represented
by the deepest blue. **b)** Comparison of the monthly average wind speed for the short-
(1980-2015; 727 stations) and long-period (1980-2019; 601 stations)

**3.3 Station relocations caused by urbanization**
Another key factor influencing wind speed measurements is the relocation of
stations. We found that there is a clear data jump caused by relocations in some of the
stations. Taking the station located in Qinghai (station ID 52974) as an example, we
detected an abrupt jump in wind speed in January 2016. This date coincides with the
relocation of the station from 35°31′N, 102°01′E (ID 52974-1) in December 2015 to
35°33′N, 102°02′E (ID 52974-2) in January 2016 (Figures 3c & 3d). The relocation is
potentially attributed to the urban growth around the station. As viewed by satellite
images from Google Earth Pro, there is a rapid urban expansion from 2006 (Figure 3a)
to 2012 (Figure 3b), especially towards the Northeast of the station, during wind speed
records also experienced a decrease (Figure 3c). A similar decrease in both daily mean
wind speed and maximum wind speed caused by urbanization was also reported in the
Yangtze River region (Zhang et al., 2022). To eliminate the effect of buildings on the
wind speed measurements, Station 52974 was moved to 4 km away from its previous
location (Figure 3d) so that wind speed is properly measured without artificial obstacles
in the surroundings.

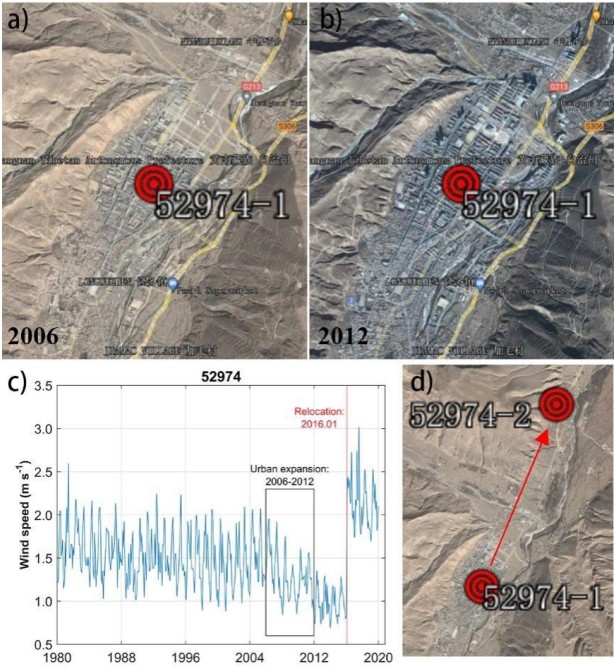


**Figure 3. Example of station relocation caused by rapid urbanization growth. a-b)**
Landsat images crop from © Google Earth near Station 52974 in 2006 and 2012,
respectively. **c)** the wind speed change with urbanization and relocation. **d)** Landsat
images of the station relocation crop from © Google Earth.


Though some stations were influenced by station relocation as shown in Figure 3,
a larger fraction (79%) of stations show no change in wind speed after the relocation.
Further checking the raw record of locations for those stations, we find that one reason
is that some "relocations" result from wrong location records. For example, Station
52974 is mistakenly detected with three relocations (Figure 4a). However, only the first
relocation is real and the latter two are results of location encoding change from 10202
(interpreted as 102°02′) to 1022 (interpreted as 10°22′) and back. Another possible
reason is that the relocation did happen but the data has been corrected. According to



the *Provisional Regulations on Relocation, Construction and Removal of National Ground Meteorological Observation* announced by China's government in 2012, station relocations should have 1-2 years of parallel observations for data correction (CMA, 2012). This process may fix some of those discontinuities but not all (Feng et al., 2004; Fu et al., 2011; Patzert et al., 2016; Tian et al., 2019; Yang et al., 2021). For example, Station 59287, the only national basic weather station in Guangzhou, experienced two relocations in both 1996 and 2011, which is confirmed by the metadata (CMA, 2011). After correction, the 1996 relocation doesn't show a sharp breaking point but the 2011 one does (Supplementary Figure S4).

To examine whether the relocation caused a substantial change in the wind speed record, we identified the most abrupt change in the wind speed time series and checked whether a relocation happened near the change point (see details in *Methods 2.3*). Out of the 432 relocated stations, 90 were deleted because the most significant shift in mean is at the time of the relocation, and hence this is the most likely cause. We then took the average of the "deleted relocation" stations and "reserved relocation" stations separately. The "deleted relocation" group shows an abnormally rapid increase in the recent two decades (Figure 4b). While the "reserved relocation" group is similar to stations without relocation (Supplementary Figure S5). To exclude the impact of different station counts in each category (fewer stations mean higher sensitivity to the individual abnormal station), we performed 300 samples using a random draw of 90 stations from the "reserved relocation" group and showed them in grey lines in Figure 4d. None of the grey lines shows an abnormal trend as the "deleted relocation" group. This proves that our method is efficient in identifying problematic stations.

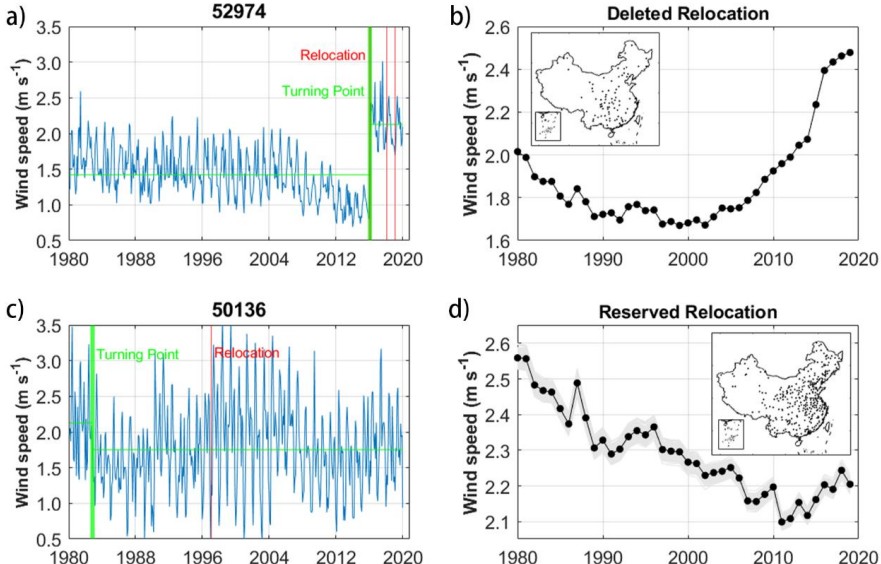

**Figure 4. Comparison of deleted relocated stations and reserved ones. a)** The wind
speed data breakpoint and relocations of one example of deleted relocation, Station
52974. **b)** The station average wind speed of 90 deleted relocated stations. The inset
shows the station distribution across China. **c)** One example of reserved relocation,
Station 50136. **d)** The station average wind speed of 342 reserved relocated stations.
The grey lines are the averages of 300 samples, each with 90 randomly drawn reserved
relocated stations. Maps information are from Department of Natural Resources
standard map service system of China.

**3.4 Average method used to calculate the national average**

In the station average time series, the breakpoint was detected in 2012 (P < 0.001)
with a trend of -0.012 m s$^{-1}$ year$^{-1}$ (R$^2$ = 0.90, P <0.001) before and +0.013 m s$^{-1}$ year$^{-1}$
(R$^2$ = 0.70, P <0.01) after (Figure 5b). The increasing trend decreased by 41% after
deleting those relocation-affected stations, compared with the +0.022 m s$^{-1}$ year$^{-1}$ in
Figure 2b (also reported by Liu et al., 2022a). But the trend is larger than the +0.011 m
s$^{-1}$ yr$^{-1}$, reported by Yang et al. (2021), with all the recorded location changed stations
deleted without checking whether the station is affected by the relocation.



We further used Thiessen Polygon (Thiessen, 1911) to give different weights to

each station according to their representing area, i.e., large weight for stations located
in sparse stations area (Figure 5a) and compare the result with the station average
(Figure 5b). Thiessen Polygon is essentially the finest divided subregion, which splits
the region into the smallest representative area and ensure there is a station in each
subregion. The Thiessen polygon weighted average is overall higher than the station
average. This can be explained by the increasing weight of stations in North West China
with higher wind speeds (Liu et al., 2019). While in the Thiessen polygon weighted
average time series, there are two breakpoints in 2000 ($P < 0.001$) and 2013 ($P < 0.01$).
The trend changes from quick decrease (-0.020 m s$^{-1}$ year$^{-1}$, $R^2 = 0.94$, $P < 0.001$) to
unstable moderate decrease (-0.004 m s$^{-1}$ year$^{-1}$, $R^2 = 0.17$, $P = 0.14$) and quickly
increase (+0.017 m s$^{-1}$ year$^{-1}$, $R^2 = 0.64$, $P < 0.05$). The increasing trend in the recent
decade increased by 31% (from +0.013 m s$^{-1}$ year$^{-1}$ to +0.017 m s$^{-1}$ year$^{-1}$) after using
the Thiessen polygon approach.

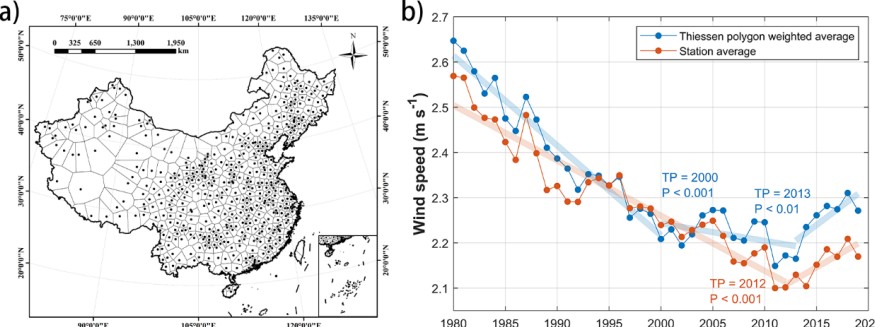


**Figure 5. Thiessen polygons and the comparison between Thiessen polygon**
**weighted average and station average. a)** The Thiessen polygon map of the 511
qualified stations. b) The comparison of station average wind speed (orange line) and
Thiessen polygon weighted average (blue line) across China for 1980-2019. The linear
fitting models are shown in translucent thick lines accordingly. Maps information are
from Department of Natural Resources standard map service system of China.

**4. Conclusions**



Continuity is crucial for meteorological observation data. However, either the
updates in the anemograph, the relocation caused by urbanization or the methods of
data logging will affect wind speed data continuity. In this study, we comprehensively
examined the discontinuity in wind speed data using a Chinese dataset. We found that
updates to the automatic anemometer improved the observation frequency and
instrument sensitivity, decreasing the zero-value daily wind speed data and increasing
data accuracy. We also propose comparing the discontinuity in time series with recorded
station relocation to check whether a relocation caused a breakpoint. We found that 90
stations were affected by the relocation and show a quickly increasing wind speed in
the recent two decades. After excluding those problematic stations, the wind speed
reversal trend is reduced by 41% but still strong ($P < 0.001$, with an increasing trend of
$+0.013$ m s$^{-1}$ year$^{-1}$). The increasing trend reaches $+0.017$ m s$^{-1}$ year$^{-1}$ ($R^2 = 0.64$, $P <$
$0.05$) after using Thiessen Polygon, which gives the stations in North West China a
larger weight because their small number but located in a large area,
Though lots of methods (Masters et al., 2010; Wan et al., 2010; Cao & Yan, 2012;
Hong et al., 2014; He et al., 2014; Azorin-Molina et al., 2018; Camuffo et al., 2020)
were proposed to handle those problems, a comprehensive summary of them is lacked.
This study fills this research niche. However, it is hard for external researchers to
provide a better solution without a collaboration with National Weather Services and
the access to station data records and/or metadata. Therefore, we hope National Weather
Services could improve the data quality based on these feedbacks and complete the
process by introducing an R package with open-source code on GitHub. This way, not
only the data is easier to get and process, but also researchers can contribute to improve
the dataset. One such example is the "rnpn" package to access and process USA
National Phenology Network data (https://github.com/usa-npn/rnpn). Anyway, all raw
data processing has limitations and adds additional uncertainty. As we keep reporting
problems in datasets and improving our processing method, we should also pay more
attention to increasing the quality and homogeneity of the wind data. This requires
raising awareness of the importance of protecting the environment around the



observation station and avoiding relocations.



**Supplemental Information**

Document S1. Supplemental Information, Table S1, Figures S1 – S5.

**Acknowledgements**

Thanking Robert Dunn (UK Met Office) for discussions and comments on the manuscript. The authors wish to acknowledge the reviewers for their detailed and helpful comments to the original manuscript.

This study was supported by the National Natural Science Foundation of China (grant no. 42071022), the Swedish Formas (2019–00509 and 2017–01408) and VR (2021–02163 and 2019–03954), and the start-up fund provided by Southern University of Science and Technology (no. 29/Y01296122). C. A-M. was supported by VENTS (GVA-AICO/2021/023), the CSIC Interdisciplinary Thematic Platform (PTI) Clima (PTI-CLIMA), the 2021 Leonardo Grant for Researchers and Cultural Creators, BBVA Foundation, and the "Unidad Asociada CSIC-Universidad de Vigo: Grupo de Física de la Atmosfera y del Océano".

**Data availability statement**

The data that support the findings of this study are available upon request from the authors.

**Author contributions**

**Zhenzhong Zeng:** Conceptualization, Methodology **Yi Liu:** Methodology, Software, Writing – Draft **Lihong Zhou:** Methodology, Data Curation **All other authors:** Writing – Review & Editing

**Declaration of interest**

The authors declare no competing financial interests.



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
