# Peer review of "Impacts of anemometer changes, site relocations and processing 1 methods on wind speed trends in China 2 Yi Liu1, Lihong Zhou1, Yingzuo Qin1, Cesar Azorin-Molina2, Cheng Shen3, Rongrong 3 Xu1\*, Zhenzhong Zeng1\* 4 <su"

_Atmospheric Measurement Techniques, 2023_

## Author Comment (AC1)

*Reviewer #1: The topic is highly relevant and supports the projects to improve climatological data quality. This is important to understand studies on trends in climate associated to climate change. Attention is given to the movement of observing sites due to urbanization in connection to harmonisation of datasets. Less attention is given to global activities, under the governance of WMO, to improve harmisation of climate datasets world wide although these activities are recommended in the conclusions. The examples provided give a good idea of the issues with harmonisation in practice, in particular for an immense area like China.*

**[Response]**

Thank you for your careful review of our manuscript " Impacts of anemometer changes, site relocations and processing methods on wind speed trends in China" and for the valuable feedback. We understand your concerns about substantial problems and have diligently worked to address the major points you raised.

We acknowledge your observation regarding the emphasis on the movement of observing sites due to urbanization and the harmonization of datasets. Your feedback has highlighted a gap in our discussion. We read more documents regarding those global activities to improve the harmonization of climate datasets under the governance of WMO and add more referees and discussions to the manuscript.

**[Reviewer #1 Major comment 1]**

*Although to topic is clear and plenty references are provided, a reference to the WMO World Climate Programme (WCP) is missing, in particular to the World Climate Data and Monitoring Programme (WCDMP) (see https://community.wmo.int/en/world-climate-programme-wcp). Within this program a number of guides are published which are relevant to learn from, like WMO/TD- No. 1186; WCDMP- No. 53 (Guidelines on climate metadata and homogenization). Also the constraint to have appropriate metadata in order to classify the quality and ability to process data is not regarded.*

**[Response]**

Thank you for your insightful feedback and recommendations. We acknowledge the oversight in not referencing the WMO World Climate Programme (WCP), particularly the World Climate Data and Monitoring Programme (WCDMP). We have now incorporated references to

the relevant guides, including WMO/TD- No. 1186 and WCDMP- No. 53 (Guidelines on climate metadata and homogenization), into our manuscript.

However, most of the metadata is not public, therefore we added emphasis of the importance of publishing metadata in the manuscript: "Also, it is hard for external researchers to provide a better solution without a collaboration with National Weather Services and the access to station data records and/or metadata. Therefore, we hope National Weather Services could improve the data quality based on these feedbacks and World Climate Data and Monitoring Programme's guidances and complete the process by introducing an R package with open-source code on GitHub and publishing the metadata." (Line 359-364 in the clean version of the revised manuscript)

**[Reviewer #1 Major comment 2]**

*In the introduction (2nd all.) surface friction is mentioned, but in the paper no attention is given to the parameter terrain roughness as defined and explained in WMO-No. 8, Vol. I (Wind Chapter). Trends in roughness are an essential analyses critria when evaluation wind datasets.*

**[Response]**

Thank you for your astute observation regarding the omission of terrain roughness in our paper, especially as it pertains to the guidelines set out in WMO-No. 8, "*Roughness around the measuring site has to be documented*" and "*any significant changes (growth of vegetation, new buildings) should be recorded in the station logbook, and dated*". While we did mention surface friction in the introduction, we recognize that we did not delve into the specifics of terrain roughness in the subsequent sections.

In light of your feedback, we have expanded our discussion on this issue. While we acknowledge the limitations of our data, which lacks detailed roughness records, we have attempted to estimate roughness changes using satellite imagery. We added more discussion about this issue: "However, this estimation of roughness change based on satellite data is rough. A more proper way as required by the World Climate Data and Monitoring Programme is to record the change in the station logbook (WMO, 2021), which will provide more reliable information about the quality of the data. But most stations don't have such a record." (Line 233-237)

To further address this gap, we've incorporated a quantitative analysis using a global roughness model. We analyzed the roughness change for the sample station in Figure 3 using the method proposed by Chappell & Webb (2016). Our findings, presented in Supplementary Figure S4, corroborate our satellite-based estimations, indicating a significant increase in roughness between 2000 and 2010. This likely influenced the observed wind speed decline, prompting the station's relocation. We have added supplementary figure S4 and further explained this in the main text: "Despite the absence of mete data, we used an established global roughness model through satellite albedo observations to monitor alterations in surface roughness. For the selected station, we employed the roughness estimation technique devised by Chappell & Webb (2016) to analyze changes in roughness across a 5 km x 5 km area encompassing the station's location. Our quantitative examination of roughness alterations aligns with the findings derived from satellite imagery analysis, affirming a pronounced increase in roughness between 2000 and 2010 (Supplementary Figure S4). This increase in roughness likely contributed to the observed decline in wind speed and ultimately compelled the relocation of the station." (Line 237-245)

[Figure]

**Figure S4. Surface roughness change around station 52974.** The roughness is evaluated by u*/Uh described in Chappell & Webb (2016).

Reference:

Chappell, A., & Webb, N. P. (2016). Using albedo to reform wind erosion modelling, mapping and monitoring. *Aeolian Research*, **23**, 63-78.

**[Reviewer #1 Major comment 3]**

*The quality of wind data is associated with siting criteria, required functional specifications of wind sensors and their maintenance policy. These topics are not refered to,but relavant because trends in these items will affect data homogenity.*

**[Response]**

Thank you for highlighting the importance of siting criteria, functional specifications of wind sensors, and their maintenance policy in relation to the quality of wind data. In our manuscript, we did touch upon this point, mentioning: "This change is potentially influenced by variations in measurement frequency, anemometer type, and data logging, as documented by Xin et al. (2012)" (Line 163-165). However, we recognize the limitations posed by the lack of comprehensive and openly accessible records on siting criteria for many stations in China. While studies like that of Xin et al. (2012) have conducted field investigations and reviewed internal documents for some stations, the majority lack publicly available records. This absence of information undoubtedly impedes efforts to enhance data quality.

In light of your feedback, we have expanded our discussion in the manuscript to emphasize the significance of documenting siting criteria, wind sensor specifications, and maintenance policies. We added that specific information that should be included in the data to the manuscript: "This example shows us the importance of recording siting criteria, required functional specifications of wind sensors and maintenance policy. However, those records are missing for most of the stations which hindered the quality classification and data processing." (Line 187-190)

**[Reviewer #1 Miner comment 1]**

*[22] reference to the WMO World Climate Programme is missing*

**[Response]**

We have added "World Meteorological Organization (WMO) World Climate Programme guidelines". (Line 22)

**[Reviewer #1 Miner comment 2]**

*[24] (~10m), to be 10 m [value and unit to be separated by a space]*

**[Response]**

We have added the space.

**[Reviewer #1 Miner comment 3]**

*[35] homogenization to become data homogenization.*

**[Response]**

We have made the change accordingly.

**[Reviewer #1 Miner comment 4]**

*[77] refer also to WMO World Climate Programme is missing, including policies on data homogenization.*

**[Response]**

We have added reference to it: "World Meteorological Organization (WMO) World Climate Programme".

**[Reviewer #1 Miner comment 5]**

*[77] missing reference to WMO, 2003: WMO/TD - No. 1186; WCDMP - No. 53, Guidelines on Climate Metadata and Homogenization, https://library.wmo.int/doc_num.php?explnum_id=10751*

**[Response]**

We have added this reference: "World Meteorological Organization (WMO, 2003). Guidelines on Climate Metadata and Homogenization. *WMO/TD - No. 1186; WCDMP - No. 53*. https://library.wmo.int/doc_num.php?explnum_id=10751".

**[Reviewer #1 Miner comment 6]**

*[90] Chosen is for the Thiessen Polygon approach (Thiessen, 1991), but explanation for this choice is not provided.*

**[Response]**

We have added explanation: "However, given station distribution and wind speed spatial variation are often inhomogeneous with larger wind but fewer station in Northwest while smaller wind but more stations in Southeast (Feng et al., 2004; Fu et al., 2011; Liu et al., 2019), the wind variation in Northwest is underrepresented because of few stations. An improved average method (area weighted average) to rearrange the weight for each station based on the area it represents is needed and Thiessen Polygon (Thiessen, 1991) is widely used in which the area is only determined by the station locations while other method like grids is sensitive to the grids chosen."

**[Reviewer #1 Miner comment 7]**

*[115] 1.85km and 1.85\*cosφ to become 1.85 km and 1.85 × cos φ [the symbol "×" should be used for multiplication, not "\*"].*

**[Response]**

We have edited them accordingly. (Line 119)

**[Reviewer #1 Miner comment 8]**

*[146] for "the frequency of zeros", "zeros" should be defined (not trivial): Stans "zero" for no data or for no wind.*

**[Response]**

We have added the definition: "zero wind speed". (Line 150)

**[Reviewer #1 Miner comment 9]**

*[151] "accuaracy" stands for a subjective expression. For a quantitative use of accuracy the word "uncertainty" shall be used, not "accuracy".*

**[Response]**

We have changed "increase in measure accuracy" to "decrease in measure uncertainty". (Line 155)

**[Reviewer #1 Miner comment 10]**

*[157] {see [151]}.*

**[Response]**

We have deleted "accuracy". (Line 161)

**[Reviewer #1 Miner comment 11]**

*[159] missing: information on maintenance, functional specifications of wind sensor (including changes) and proven traceability to SI (calibration, see also WMO-No. 8). Also the required siting specifications are missing (including changes), see WMO-No. 8, Vol. I.*

**[Response]**

We have expanded our discussion in the manuscript to underscore the significance of these details. We've added: "This example shows us the importance of recording siting criteria, required functional specifications of wind sensors and maintenance policy. However, those records are missing for most of the stations which hindered the quality classification and data processing." (Line 187-190)

**[Reviewer #1 Miner comment 12]**

*[183] missing: roughness data and their trend. Not determined? Are necessary to understand further analyses of data (see WMO-No. 8, Vol. I, Wind Chapter).*

**[Response]**

We fully acknowledge the importance of documenting roughness around the measuring site. As stipulated in WMO-No. 8, significant changes, such as the growth of vegetation or the construction of new buildings, should indeed be recorded and dated in the station logbook. Regrettably, our dataset lacks this specific information.

We have resorted to satellite imagery to provide an approximate estimation of roughness changes, particularly noting an increase in roughness due to the construction of more buildings around the station. However, we recognize that this method offers only a rough estimation. We've expanded our discussion in the manuscript to emphasize this limitation: "Despite the absence of mete data, we used an established global roughness model through satellite albedo observations to monitor alterations in surface roughness. For the selected station, we employed the roughness estimation technique devised by Chappell & Webb (2016) to analyze changes in roughness across a 5 km x 5 km area encompassing the station's location. Our quantitative examination of roughness alterations aligns with the findings derived from satellite imagery analysis, affirming a pronounced increase in roughness between 2000 and 2010 (Supplementary

Figure S4). This increase in roughness likely contributed to the observed decline in wind speed and ultimately compelled the relocation of the station." (Line 237-245)

**[Reviewer #1 Miner comment 13]**

*[320] sentence ends with a comma, not a dot.*

**[Response]**

We have changed the comma with a dot. (Line 355)

**[Reviewer #1 Miner comment 14]**

*[323] "lacked"; preference for "missing".*

**[Response]**

We have replaced "lacked" with "missing". (Line 358)

**[Reviewer #1 Miner comment 15]**

*[328] Refrence to WMO Worls Climate Programme, in particular the World Climate Data and Monitoring Programme stimulating the proposed actions.*

**[Response]**

We added the reference to the World Climate Data and Monitoring Programme: "Therefore, we hope National Weather Services could improve the data quality based on these feedbacks and World Climate Data and Monitoring Programme's guidances, and complete the process by introducing an R package with open-source code on GitHub and publishing the metadata" (Line 361-364)

**[Reviewer #1 Miner comment 16]**

*[476] Not a "WMO Technical Report", but a WMO Guide; to become WMO-No. 1203. Refrence to be replaced by principal source, https://library.wmo.int/doc_num.php?explnum_id=4166.*

**[Response]**

We have edited the reference accordingly: "World Meteorological Organization (WMO, 2017). WMO guidelines on the calculation of climate normals. *WMO-No. 1203*. https://library.wmo.int/doc_num.php?explnum_id=4166" (Line 518-520)

**[Reviewer #1 Miner comment 17]**

*[479] Not a "WMO Technical Report, but a WMO Guide; to become WMO-No. 1245..*

**[Response]**

We have edited the reference accordingly. (Line 521-522)

---

## Author Comment (AC2)

*Reviewer #2: The meteorological data continuity is crucial for climate and climate change areas. The paper focuses on the discontinuity of the wind speed datasets across China and quantifies the possible impacts of anemograph changes, data logging methods, and site relocations on long-term wind speed trends. The authors found that the use of advanced anemographs can increase the observation frequency and improve the instrument sensitivity, thereby enhancing data accuracy. Additionally, the paper checks station relocations by examining the altitude, latitude, and longitude information of all observation stations, identifying approximately 90 stations in China that are affected by relocation and exhibit an excessive increase in wind speed trends. Based on the discontinuity analysis, problematic stations are recognized and excluded. The wind speed data series is reconstructed and a new wind speed trend is also given in this paper.*

*The topic of the paper is suitable for Atmospheric Measurement Techniques. The authors' careful and comprehensive work in examining the wind data series across China is appreciated, and I also agree with their call for the improvement of data quality based on user feedback.*

*This paper requires some minor revisions before publication.*

**[Response]**

Thank you for your time and valuable input. We appreciate your recognition of the importance of our study on wind speed data continuity in climate research. We are encouraged by your positive comments regarding our analysis of wind speed dataset discontinuity, impacts of instrumentation changes, and site relocations. We share your commitment to improving data quality through user feedback collaboration. We have addressed the minor revisions you've suggested.

**[Reviewer #1 Major comment 1]**
*Line 101: To maintain consistency in Table S1, the "(3)" should be added before the third criterion.*
**[Response]**

Thank you for your comment. We have added "(3)" before the third criterion in the main text: "(1) <11 missing daily values in a month; (2) <5 consecutive missing daily values in a month; (3) Complete monthly values for every month during the study period." (Line 106-108 in the clean version of the revised manuscript)

**[Reviewer #1 Major comment 2]**

*Line 169: What do the "EL" and "EC" stand for?*

**[Response]**

The expression of "EL" and "EC" are only seen in Chinese literatures. Even though they are widely used in research papers and instrument documentations, we didn't find any explicit expression about what "EL" and "EC" stands for. From reading those literatures, we think "EL" possibly refers to "Electric Logging" which describe how this type of anemometer record wind speed: logging the times of electronic contact (200 meters rotation distance per contact) in 10 mins in a paper. "EC" refers to "Electronic Code" which means it use Grey Code to record more accurate wind speed. (Hu et al., 2009; Xin et al., 2012; Yang 1986).

Reference:

Hu, W., Kong, L., Zhu, X., & Xue, W. (2009). Accurancy analysis on contact anemometer self – recording records digitization processing system. *Jounal of Arid Meteorology*, **27**, 168-171. [In Chinese]

Xin, Y., Chen, H., & Li, Y. (2012). Homogeneity adjustment of annual mean wind speed and elementary calculation of fundamental wind pressure over Xinjiang meteorological stations. *Climatic and Environmental Research*, **17**(2), 184-196. [In Chinese]

Yang, Jihua. (1986). The repair and maintenance of EL anemometer. *Meteorology of Xinjiang*, **8**, 46-47. [In Chinese]

**[Reviewer #1 Major comment 3]**

*Line 284: More detailed information about the "Thiessen Polygon" method should be given.*

**[Response]**

Thank you for your suggestions. We added: "The Thiessen Polygon method, also known as the Voronoi Diagram, is a spatial analysis technique often employed in hydrology and climatology. It involves tessellating a region into polygons based on point data, such that each polygon encompasses only one data point, and every location within a polygon is closer to its associated point than any other. This method is particularly useful for interpolating values across a region when the exact nature of change between points is unknown or when changes are

abrupt. By drawing perpendicular bisectors between adjacent data points, the entire area is divided, with each polygon assuming the value of its associated data point. While straightforward and clear in its delineation, the Thiessen Polygon method assumes uniform variation within each polygon." (Line 306-315) to further explain the "Thiessen Polygon" method.

**[Reviewer #1 Major comment 4]**

*Line 285-287: Large weights of the wind speed were given in Northwestern China and the Tibetan Plateau, but these regions are also complex terrains compared to Eastern China. How does the author consider the problem of station representation?*

**[Response]**

Thank you for highlighting the complexities of terrain, especially in Northwestern China and the Tibetan Plateau, and its potential impact on station representation.

In our study, we've operated under the assumption that the wind speed at a given location is best represented by its nearest station, an approach we believe offers improvement over previous direct averaging and grid methods. However, we acknowledge the limitations of this method, especially in regions with complex terrains where both driving forces (air motion) and resisting forces (friction and terrain) introduce intricate wind variability.

While it's feasible to develop a model that integrates these factors for a more nuanced interpolation, the inherent uncertainties due to limited data availability remain a challenge. For regions with intricate terrains like Northwestern China and the Tibetan Plateau, a more robust interpolation model would benefit from increased observational data.

We have incorporated this discussion into our manuscript, emphasizing the need for enhanced observational data in complex terrains to refine interpolation models and better capture the spatial variability of wind speed: "Despite the Thiessen polygon approach already utilizing the nearest station observation to represent wind speed in locations lacking direct observations, it remains unsatisfactory due to the intricate spatial variability of wind speed attributed to complex terrains. To enhance the accuracy of wind speed interpolation, a more comprehensive model necessitates additional observations within areas characterized by complex terrain." (Line 326-331)

**[Reviewer #1 Major comment 5]**

*Line 295-297: The phenomenon of the increasing trend in wind speed across China in recent decades after using the Thiessen Polygon should be further explained.*

*It is suggested that the author DO MORE WORK TO CLARIFY the wind speed trends in different regions of China, at least over eastern China.*

**[Response]**

We appreciate your feedback on the observed increasing trend in wind speed across China. Indeed, even without employing the Thiessen Polygon, an increasing trend in wind speed is evident (as seen in Figure 5b), albeit less pronounced. The differentiation in trends across various regions of China, particularly the underrepresentation of areas like the North West and South West due to sparser station distribution, plays a significant role in this observation. When we transitioned to an area-based average rather than a station count-based one, the influence of regions like the North West and South West became more pronounced, given their strong increasing wind speed trends.

To address your suggestion, we've incorporated a regional analysis, presented in supplementary figure 7 and discussed in the main text: "This is because the weights of stations in North West and South West are increased when calculating the average and those area has strong increasing wind speed trend (Figure S7)." (Line 324-326)

[Figure]

**Figure S7. Regional wind speed trends.** The regional division is consistent with Figure 2a in Liu et al 2022.

---

## Author Response (AR2)

**[Comment]**

*The use of 'EC' and 'EL' is still not clarified in the revised verison. Please figure out what they are in the final version.*

**[Response]**

Thank you for the comments. In the revised manuscript, we have included the full expressions of "EL" as "Electric Logging" and "EC" as "Electric Coding." We have also added further explanation regarding the differences between these two types of anemographs for better clarity: "Then in 2005, the EL (Electric Logging) contact anemograph (Yang, 1986; Jin, 2011; Xin et al., 2012, Zhang et al., 2020), which required manual recording, was replaced by the EC (Electric Coding) photoelectric encoder self-recording type (Kuang, 2016; Jin, 2011; Xin et al., 2012). Both EL and EC types of anemographs use cup anemometers to measure wind speed. However, the EL type measures the times of electronic contact (e.g., 200 meters rotation distance per contact) in a time period, resulting in discrete records, while the EC type uses the Grey Code encoder rotating with the cup anemometers to obtain a more precise wind speed record." (Line 170-178 in the tracked version).